# High-Pressure Processing Influences Antibiotic Resistance Gene Transfer in *Listeria monocytogenes* Isolated from Food and Processing Environments

**DOI:** 10.3390/ijms252312964

**Published:** 2024-12-02

**Authors:** Patryk Wiśniewski, Wioleta Chajęcka-Wierzchowska, Anna Zadernowska

**Affiliations:** Department of Food Microbiology, Meat Technology and Chemistry, Faculty of Food Science, University of Warmia and Mazury, Plac Cieszyński 1, 10-726 Olsztyn, Poland; patryk.wisniewski@uwm.edu.pl (P.W.); wioleta.chajecka@uwm.edu.pl (W.C.-W.)

**Keywords:** antibiotic resistance genes, genes transfer, high-pressure processing

## Abstract

The study aimed to assess the high-pressure processing (HPP) impact on antibiotic resistance gene transfer in *L. monocytogenes* from food and food processing environments, both in vitro (in microbiological medium) and in situ (in carrot juice), using the membrane filter method. Survival, recovery, and frequency of antibiotic resistance gene transfer analyses were performed by treating samples with HPP at different pressures (200 MPa and 400 MPa). The results showed that the higher pressure (400 MPa) had a significant effect on increasing the transfer frequency of genes such as *fosX*, encoding fosfomycin resistance, and *tet_A1*, *tet_A3*, *tetC*, responsible for tetracycline resistance, both in vitro and in situ. In contrast, the *Lde* gene (the gene encoding ciprofloxacin resistance) was not transferred under any conditions. In the food matrix (carrot juice), greater variability in results was observed, suggesting that food matrices may have a protective effect on bacteria and modify HPP efficacy. In general, an increase in MIC values for antibiotics was noted in transconjugants compared to donors. Genotypic analysis of transconjugants showed differences in genetic structure, especially after exposure to 400 MPa pressure, indicating genotypic changes induced by pressure stress. The study confirms the possibility of antibiotic resistance genes transfer in the food environment, even from strains showing initial susceptibility to antibiotics carrying so-called silent antibiotic resistance genes, highlighting the public health risk of the potential spread of antibiotic-resistant strains through the food chain. The findings suggest that high-pressure processing can increase and decrease the frequency of resistance gene transfer depending on the strain, antibiotic combination, and processing conditions.

## 1. Introduction

The growing concern over antibiotic resistance among foodborne pathogens is a critical public health issue. *Listeria monocytogenes*, the bacterium responsible for causing listeriosis, is particularly worrying due to its high mortality rate and ability to survive in harsh food processing environments. *L. monocytogenes* is exceptionally resilient to harsh environmental conditions, making it particularly challenging in the food industry. Although HPP effectively eliminates many microorganisms, *L. monocytogenes* shows resistance under certain conditions, especially in the presence of biofilms or when previously exposed to sublethal environmental stresses [1]. Additionally, *L. monocytogenes* can enter the food chain through contaminated soil, water, and processing equipment, leading to contamination of fresh produce, such as vegetables used in juices, and ready-to-eat food products [2]. Contamination of the food chain by *L. monocytogenes* poses a significant challenge for the industry, requiring not only stringent hygiene protocols but also improved control strategies to reduce the risk of infections [3].

The emergence of antibiotic-resistant strains of *L. monocytogenes* exacerbates this threat, complicating infection treatment, particularly in susceptible populations, such as pregnant women, the elderly, and the immunocompromised [4,5]. *L. monocytogenes* is also characterized by its ability to resist multiple antibiotics, further complicating its control and treatment of infection [6]. Listeriosis has a high mortality rate, especially in immunocompromised individuals. According to the European Food Safety Authority (EFSA), *L. monocytogenes* caused the highest rates of hospitalization and mortality among zoonotic diseases in Europe, with a reported incidence of 0.44 cases per 100,000 people in 2021 and 1.7 cases per 100,000 people over the age of 64 [7]. The increasing frequency of zoonotic outbreaks and the persistence of resistant strains underscore the urgent need for comprehensive monitoring and improved food safety measures throughout the production chain.

Antibiotic resistance in *L. monocytogenes* is mainly due to horizontal gene transfer (HGT), which allows the bacteria to acquire genetic material from other microorganisms in the host environment. This process is a major factor in spreading antibiotic resistance in natural environments and the food industry [7]. The transfer of antibiotic resistance genes (ARGs) in *L. monocytogenes* strains is observed in various studies [5,6,7,8]. *L. monocytogenes* has been shown to acquire resistance through plasmids and transposons that can be transferred between different bacterial species, including those found in food products and the human microflora [5,9].

Interestingly, resistance gene transfer is not limited to intra-species exchange but also occurs between different microorganisms, such as *Enterococcus faecalis* to *L. monocytogenes* [5,10]. Recent studies have shown that *L. monocytogenes* can acquire resistance genes from other bacteria, such as *E. faecium*, often found in fermented meat products [5]. The transfer of tetracycline resistance via conjugation with enterococci highlights the potential for spreading resistance genes in the food environment [5]. Furthermore, studies have also shown that resistance genes can be transferred between bacteria in vitro and in vivo, posing a significant challenge to public health and food safety [11]. Moreover, the presence of resistance genes in *Listeria* strains isolated from food and clinical samples indicates the potential for these genes to spread through the food chain, posing a risk to the effective treatment of listeriosis [12,13,14,15].

Of particular concern in the food processing context is that high-pressure processing (HPP) can potentially affect the activation and expression of resistance genes, increasing the risk of their further spread [16]. In recent years, HPP has gained popularity as a non-thermal method of food preservation, capable of inactivating pathogens while maintaining the sensory and nutritional quality of food products [4]. The HPP involves the application of high pressure (100–600 MPa) for several minutes at room temperature or lower, which usually leads to damage to cell structures and subsequent inactivation of the microorganisms. Although HPP is effective in reducing the number of *L. monocytogenes* to below-detectable levels, the effect of this technology on the transfer and expression of antibiotic-resistance genes remains the subject of intensive research [16]. In the context of food safety, HPP is considered an effective method of reducing pathogen abundance to levels that are safe for consumers. However, concerns have been raised regarding the potential recovery of bacterial populations following treatment and the persistence of phenotypic and genotypic changes induced by pressure stress [4]. In addition, research suggests that HPP may lead to a sustained increase in antibiotic resistance, posing additional challenges for the food industry and health systems [16].

Several studies have suggested that HPP treatment may lead to stressed cellular responses in microorganisms that may affect the expression of virulence factors and antibiotic-resistance genes [17,18,19]. For example, it has been observed that after HPP treatment, the frequency of transfer of certain resistance genes, such as those encoding resistance to tetracyclines, ampicillin, and chloramphenicol, increases both in vitro and in situ in a food matrix in strains form starter cultures [18]. In a study on the effect of HPP on antibiotic-susceptibility, it was observed that some strains of *L. monocytogenes* that were initially susceptible to lincomycin, fosfomycin, trimethoprim/sulfamethoxazole, and tetracycline became resistant to these antibiotics after HPP treatment. This phenomenon was associated with an increase in minimum inhibitory concentrations (MICs) and increased expression of antibiotic resistance genes, further complicating the management of antibiotic resistance in food processing environments [16]. Studies have shown that *L. monocytogenes* isolates from various sources, including food and clinical settings, often carry multiple resistance genes, contributing to their resistance to antimicrobial treatment [16,20,21,22,23,24]. Furthermore, the prevalence of resistance genes documented in large-scale studies highlights the need for continuous monitoring and understanding of these dynamics [7].

In addition, HPP may also affect resistance gene transfer in the environment. In studies on an in vivo model, which better reflects the actual conditions in living organisms, it was found that the level of resistance gene transfer can be higher than under in vitro laboratory conditions. This indicates the need for more advanced models investigating resistance gene transfer to understand better and control the spread of antibiotic resistance [11]. Research into the impact of HPP on antibiotic resistance gene transfer in *L. monocytogenes* is key to understanding the risks associated with this technology in the food industry. Therefore, the study aimed to assess HPP treatment’s impact on antibiotic resistance gene transfer in *L. monocytogenes*, both in vitro and in situ, using isolates from food and food processing environments. The study aimed to understand if and how HPP can affect gene transfer mechanisms, thereby influencing the spread of antibiotic resistance in food-borne pathogens.

## 2. Results

### 2.1. Donor Isolates Characterization

At the beginning of the study, all *L. monocytogenes* strains used as donors were tested for plasmid localization of antibiotic-resistance genes. Genes encoding resistance to a particular antibiotic and plasmid-localized were screened for transferability. The characteristics of isolates carrying specific genes on plasmids are presented in Table 1.

### 2.2. Survival and Recovery Analysis

The survival analysis was carried out using plate count methods in two experimental variants—in vitro in BHI microbiological medium (Merck, Darmstadt, Germany) and in situ in a food matrix, carrot juice (Vital Fresh, Łęczyca, Poland). As in the authors’ previous study [16], the lower pressure variant did not reduce the live cell counts of the individual strains, compared to the initial count regardless of the experimental variant. Higher pressure reduced abundance to below the detection limit (<10 CFU/mL) for isolates analyzed in vitro (Table 2). For isolates pressurized in carrot juice, three isolates (Lm_4, Lm_5, and Lm_6) were detectable using the counting method. The counts of the individual strains were 2 × 10^1^ CFU/mL, 7 × 10^1^ CFU/mL, and 3 × 10^1^ CFU/mL, respectively (Table 3). The remaining strains were subjected to recovery in carrot juice.

All strains obtained, both after treatment with 200 MPa (in medium and food matrix) and recovered after treatment with 400 MPa (in medium and food matrix), were immediately subjected to analyses of the possibility of transfer of antibiotic resistance genes.

### 2.3. Antibiotic Resistance Genes Transfer

#### 2.3.1. Gene Transfer Frequency

Six *L. monocytogenes* strains were used as donors, while *Enterococcus faecalis* strain JH2-2 served as the recipient in in vitro and in situ gene transfer experiments. Table 4 and Appendix A present the results of testing the effect of HPP on the frequency of antibiotic resistance gene transfer in the growth medium (in vitro) and the food matrix (in situ) (see Appendix A for more details).Evaluation of the effect of HPP revealed that its impact varied significantly depending on the applied pressure and the experimental environment. Higher pressure values, particularly 400 MPa, were associated with an increased frequency of gene transfer, especially in the food matrix, while the growth medium demonstrated more stable results.

For tetracycline (TET), a substantial increase in gene transfer frequency was observed in transconjugant TC-Lm_1 after recovery following exposure to 400 MPa in the food matrix, where the value reached 1.51 ± 0.03 × 10^2^, compared to the control value of 4.91 ± 0.07 × 10^−1^. Similarly, increases were noted for TC-Lm_4 (0.41 ± 0.02 × 10^1^) and TC-Lm_6 (0.52 ± 0.01 × 10^1^) in the food matrix following 400 MPa. In contrast, the growth medium exhibited smaller differences, such as in TC-Lm_4, where the frequency reached 3.40 ± 0.11 × 10^−1^ post-400 MPa, compared to the control value of 1.29 ± 0.02 × 10^−1^. For lincomycin (LIN), similar patterns were observed. In the food matrix, TC-Lm_1 showed a frequency of 0.18 ± 0.01 × 10^1^ after 400 MPa, more than threefold higher than the control value of 4.79 ± 0.04 × 10^−1^. TC-Lm_4 also demonstrated an increase in the food matrix (0.19 ± 0.01 × 10^1^ post-400 MPa), while the growth medium revealed only modest changes, with values such as 0.10 ± 0.00 × 10^1^ for TC-Lm_4, compared to 3.19 ± 0.03 × 10^−1^ in the control.

For ciprofloxacin (CIP), an increase in gene transfer frequency was noted in the food matrix after 400 MPa, particularly in TC-Lm_1, where the value soared to 8.51 ± 0.06 × 10^1^, over twenty times higher than the control (2.56 ± 0.03 × 10^−1^). Similarly, TC-Lm_6 displayed a frequency of 5.65 ± 0.21 × 10^2^, compared to the control value of 1.81 ± 0.01 × 10^−2^. In the growth medium, changes were less pronounced; for instance, TC-Lm_2 exhibited a frequency of 6.76 ± 0.30 × 10^−1^ after 400 MPa, compared to the control value of 0.26 ± 0.00 × 10^1^. Fosfomycin (FOS) and clindamycin (DA) exhibited smaller increases in the food matrix. For example, TC-Lm_5 showed a frequency of 7.17 ± 0.18 × 10^−1^ after 400 MPa, slightly higher than the control (1.46 ± 0.04 × 10^−1^). For clindamycin, TC-Lm_4 demonstrated a frequency of 0.16 ± 0.00 × 10^1^ following 400 MPa in the food matrix, compared to the control value of 5.12 ± 0.04 × 10^−1^.

Conversely, some transconjugants showed decreases in gene transfer frequency after HPP. For TET, TC-Lm_3 displayed a marked reduction in the food matrix, where the frequency dropped to 0.70 ± 0.02 × 10^−1^ after 400 MPa, compared to the control (2.30 ± 0.09 × 10^−1^). In the growth medium, the frequency for TC-Lm_3 decreased to 0.16 ± 0.00 × 10^1^, compared to the control value of 3.53 ± 0.03 × 10^−1^. Similarly, TC-Lm_5 in the food matrix exhibited a frequency of 0.13 ± 0.01 × 10^1^ after 400 MPa, compared to the control (4.71 ± 0.07 × 10^−1^). For LIN, decreases were noted in TC-Lm_2 and TC-Lm_3. In the growth medium, TC-Lm_2 showed a frequency of 3.51 ± 0.07 × 10^−1^ following 400 MPa, compared to 4.01 ± 0.01 × 10^−1^ in the control, while TC-Lm_3 in the food matrix demonstrated a reduction to 3.38 ± 0.07 × 10^−1^, compared to 2.17 ± 0.07 × 10^−1^ in the control.

For CIP, TC-Lm_4 in the growth medium exhibited a decrease in frequency to 1.51 ± 0.04 × 10^−1^ post-400 MPa, compared to 6.15 ± 0.05 × 10^−1^ in the control. In TC-Lm_5, the frequency in the food matrix dropped significantly to 5.99 ± 0.01 × 10^1^ after 400 MPa, compared to the control value of 7.66 ± 0.03 × 10^−2^. Similar decreases were observed for FOS and DA. For FOS, TC-Lm_3 in the food matrix showed a reduction to 4.87 ± 0.04 × 10^−1^ following 400 MPa, compared to 0.45 ± 0.00 × 10^1^ in the control. TC-Lm_4 showed a similar trend, with a frequency of 0.98 ± 0.02 × 10^1^ after 400 MPa, compared to 5.64 ± 0.05 × 10^−1^ in the control. For DA, TC-Lm_5 in the growth medium exhibited a frequency of 8.53 ± 0.19 × 10^−1^ post-400 MPa, compared to 1.92 ± 0.02 × 10^−1^ in the control. Similarly, TC-Lm_6 in the food matrix showed a reduction to 8.22 ± 0.05 × 10^−1^, compared to 6.87 ± 0.05 × 10^−1^ in the control.

These results highlight the complexity of HPP’s impact on antibiotic resistance gene transfer frequencies. Depending on the pressure level, environment, and specific transconjugants, HPP at 400 MPa may either enhance or inhibit gene transfer. Statistical analysis (Appendix A) using Friedman’s ANOVA, Kruskal–Wallis tests, and Spearman correlation revealed significant differences and relationships between pressure conditions, antibiotic types, and the frequency of resistance gene transfer. Friedman’s ANOVA (Appendix A) showed notable differences for ciprofloxacin under in situ conditions (*p* = 0.002479), while lincomycin and tetracycline demonstrated trends under varying conditions. Appendix A highlights the impact of pressure on gene transfer, showing significant effects for lincomycin at 200 MPa (*p* = 0.036032) and for ciprofloxacin and fosfomycin after recovery from 400 MPa in situ (*p* = 0.036032). Kruskal–Wallis tests (Appendix A) confirmed significant differences in gene transfer frequency among antibiotics after 400 MPa in situ (*p* = 0.003999). Lastly, Spearman correlation analysis (Appendix A) revealed a strong relationship between HPP and gene transfer frequencies, particularly for fosfomycin at 200 MPa (correlation coefficient 0.787172). These findings underscore the intricate interplay of environmental conditions and pressure levels on the modulation of antibiotic resistance gene transfer during HPP.

#### 2.3.2. Possibility and Confirmation of Gene Transfer

In the transconjugants obtained after gene transfer, the presence of genes encoding antibiotic resistance in both genomic and plasmid DNA was determined (Table 5). The results for tetracycline indicate the possibility of transfer of all three tetracycline resistance genes analyzed (*tet_A1*, *tet_A3*, *tetC*), due to the observed presence of transconjugants in genomic and/or plasmid DNA, with all three genes detected in transconjugants not exposed to HPP and after exposure to 400 MPa in vitro. The *tetC* gene was found in control (in vitro), as well as after exposure to 400 MPa (in vitro). No transfer of this gene was observed in situ indicating that the gene can only be transferred under certain conditions. In the case of lincomycin, only the absence of *lin* gene transfer was observed when higher HPP process conditions were applied in vitro. In the other conditions analyzed, the *lin* gene was present in both genomic and plasmid DNA independently of the transconjugant. In the case of ciprofloxacin, the resistance gene for this antibiotic (*Lde*) was not found in any of the transconjugants tested, indicating that this gene could not be transferred. The *fosX* gene (encoding fosfomycin resistance) was detected in the plasmid DNA of transconjugants not treated with HPP and after 200 MPa (both in vitro and in situ) in five of the six transconjugants analyzed. The gene was also present in both genomic and plasmid DNA in transconjugants after only 400 MPa treatment in vitro. The potential for transfer of the gene encoding clindamycin resistance (*lnuA*) was also found only in transconjugants not exposed to HPP under both in vitro and in situ conditions—the gene was present in the plasmid DNA of the transconjugants. The *lnuA* gene was not found in transconjugants after exposure to 200 MPa and 400 MPa, indicating the impossibility of gene transfer from the donor to the recipient.

#### 2.3.3. Changes in the Values of Minimum Inhibitory Concentrations (MICs) in the Obtained Transconjugants

The obtained transconjugants were tested for MIC values (Table 6). MIC values were from 0.5 to 2 μg/mL for tetracycline, 2–256 μg/mL for lincomycin, 0.25–2 μg/mL for ciprofloxacin, 128–256 μg/mL for fosfomycin and 1–2 μg/mL for clindamycin. The values obtained were generally higher than in the donors, with a few cases of a decrease in MIC for lincomycin TC-Lm_2 200 MPa (in situ), TC-Lm_2 200 MPa (in vitro), TC-Lm_1 200 MPa (in vitro), TC-Lm_1 200 MPa (in situ)), fosfomycin (TC-Lm_2 200 MPa (in situ), TC-Lm_3 200 MPa (in situ), TC-Lm_2 200 MPa (in vitro), TC-Lm_2 400 MPa_recovery (in vitro), TC-Lm_1 Control (in situ), TC-Lm_2 Control (in situ), TC-Lm_2 400 MPa_recovery MPa (in vitro)) and clindamycin (TC-Lm_4 da 200 MPa (in situ), TC-Lm_1 200 MPa (in vitro), TC-Lm_1 400 MPa_recovery MPa (in vitro), TC-Lm_5 Control (in vitro), TC-Lm_5 Control (in situ)) and several cases of unrecorded changes in MIC values for fosfomycin (TC-Lm_2 200 MPa (in situ), TC-Lm_1 200 MPa (in situ), TC-Lm_1 200 MPa (in vitro), TC-Lm_1 Control (in vitro), TC-Lm_1 400 MPa_recovery (in vitro)), tetracycline (TC-Lm_5 200 MPa (in situ)) and clindamycin (TC-Lm_3 400 MPa_recovery (in situ), TC-Lm_1 Control (in vitro), TC-Lm_1 Control (in situ)). There was an increase in MIC values in all casesm compared to the donors for ciprofloxacin. These results show that the transconjugants had antibiotic-resistance genes that were expressed.

## 3. Discussion

*L. monocytogenes* is known for its high resistance to environmental stresses, including high pressure, allowing it to survive in various environments, especially food. Studies indicate that the survival of bacteria, such as *L. monocytogenes*, may vary significantly depending on the food matrix, which may exhibit protective properties against the microorganisms [25]. In our study, the effect of pressure on bacterial survival differed between in vitro and in situ conditions in the food matrix, which was carrot juice. This aligns with the existing literature, which suggests that bacterial survival under physico-chemical stress depends on environmental factors such as substrate and stress intensity [25]. Our statistical analysis supports this observation by showing that environmental factors, such as the food matrix and pressure levels, significantly influence bacterial survival and the frequency of gene transfer under HPP conditions. For instance, Friedman’s ANOVA highlighted the effect of in situ conditions on ciprofloxacin resistance transfer frequencies (*p* = 0.002479), suggesting that food matrices like carrot juice provide a protective environment mitigating pressure-induced stress.

In our study, higher pressure reduced bacterial numbers below the limits of detection in vitro, indicating that extreme conditions in simple media can lead to bacterial death. This finding aligns with research highlighting the need for stress-resistance mechanisms under such conditions [26]. On the other hand, isolates exposed to high pressure in carrot juice showed greater resistance. Some strains were still detectable, supporting the hypothesis that food matrices may provide a protective effect by mitigating the intensity of external stressors [27]. *L. monocytogenes* can adapt to external stresses, including high pressure, through various defense mechanisms. In fermented products, the bacteria may resist adverse environmental conditions more than in simple microbial media [26]. In studies on foods such as carrot juice, it has been found that food components can promote the survival of *L. monocytogenes*, which may explain the higher CFU/mL values for some strains in our study [28].

The presence of virulent and antibiotic-resistant strains in food and food processing environments suggests the possibility of resistance gene transfer through the food chain [20,29]. Transfer of antibiotic resistance genes in *L. monocytogenes* strains is possible, as confirmed by our study and many other previous studies, and represents a significant public health risk, as it can lead to strains difficult to treat with conventional antibiotics [5,8,18,19,30].

The results on the effect of HPP on the transfer of antibiotic resistance genes between *L. monocytogenes* and *E. faecalis* strains show varying effects depending on the experimental conditions (in vitro vs. in situ) and pressure values. This variability was quantitatively supported by our findings, which revealed significant differences in gene transfer frequency among antibiotics after exposure to 400 MPa in situ (*p* = 0.003999). Furthermore, Spearman correlation analysis demonstrated a strong relationship between pressure levels and gene transfer frequency for fosfomycin resistance at 200 MPa (correlation coefficient 0.787172). Our study also confirmed the possibility that transfer of tetracycline resistance genes (*tet_A1*, *tet_A3*, *tetC*) depended on the environmental conditions of the study—all three genes were detected in transconjugants unexposed to HPP and after exposure to 400 MPa in vitro. No transfer of the *tetC* gene was detected after exposure to 200 MPa (in vitro). A similar effect was observed in in situ experiments—the *tetC* gene was not found in any of the transconjugants analyzed, indicating that transfer of this gene was not possible. Also, our previous study [19] suggests the possibility of transfer of so-called silent antibiotic resistance genes encoding resistance to: tetracyclines (*tet*(M), *tet*(L), *tet*(K), *tet*(W), and *tet*(O)) by horizontal gene transfer in both untreated and HPP-exposed strains. A study by Zarzecka et al. (2022) [31] confirms the possibility of horizontal gene transfer between bacterial strains from starter and guard cultures used in food fermentation. Transferability analysis of resistance genes showed that the strains analyzed can horizontally transfer tetracycline resistance genes *tetM* and *tetK*. It is important to observe that, in our study, strains showing initial sensitivity to antibiotics carrying so-called silent antibiotic resistance genes were also capable of gene transfer. Similarly, there was a *fosX* gene (encoding fosfomycin resistance)—transfer was higher at higher pressure values (400 MPa), compared to control and low-pressure samples (200 MPa). However, the effect of HPP was not the same for all the antibiotics analyzed—for some, such as ciprofloxacin, no transfer of the *Lde* gene encoding resistance to this antibiotic was observed under any conditions. In the case of lincomycin, transfer of *lin* gene (encoding resistance to this antibiotic) was only observed in samples not treated with HPP. The scientific literature indicates similar observations, suggesting that environmental stresses, such as high pressure, may affect the process of horizontal gene transfer [5,8,18,19,32,33,34]. There are suggestions that resistance genes to some antibiotics may be more difficult to transfer under environmental stress [35], and the failure of gene transfer may also be because, in some strains, resistance genes may reside on non-conjugative plasmids that are incapable of transfer by conjugation [32].

Nevertheless, Jahan and Holley (2016) [5] showed that *Listeria* strains can acquire tetracycline and streptomycin resistance through non-plasmid conjugation with *E. faecium* isolated from fermented sausage. This suggests that natural transfer of antibiotic resistance to *Listeria* strains may occur in the food environment, raising concerns about the safety of enterococci strains in food. Similarly, Haubert et al. (2018) [8] report that two isolates carried antibiotic resistance genes, with one carrying the *tetM* resistance gene on a plasmid, indicating potential transfer between bacteria in the food environment. Godziszewska et al. (2016) [30] confirm that food can be a reservoir of antibiotic resistance genes, with the possibility of horizontal gene transfer between different bacterial species.

The effect of HPP on antibiotic resistance gene transfer is complex and dependent on the specific environmental conditions and the type of pressure applied. While under in vitro conditions higher pressure (400 MPa) generally reduced the frequency of gene transfer, under in situ conditions the bacterial response was more complex and included both an increase and decrease in the rate of transfer depending on the combination of bacterial strain and antibiotic. This suggests the need for further research in the context of real food matrices, where environmental factors may significantly affect the effectiveness of HPPs in reducing the spread of antibiotic resistance [34]. Our results are consistent with previous reports, which indicate that moderate pressure (200 MPa) induces only minor genotypic changes, while high pressure (400 MPa) leads to more significant transformations of genetic material. Our results largely agree with the literature on the effects of pressure stress on the bacterial genome, which emphasizes that higher pressure levels induce increasingly intense adaptive changes in bacteria [36,37]. Studies show that HPP can indeed affect ARG transfer, with different pressure levels altering the frequency of conjugation. Zarzecka et al. (2023) [18] found that HPP increased the frequency of transfer of tetracycline and chloramphenicol resistance genes both in vitro and in food matrices, like the observation of increased transfer at 400 MPa pressure in our study. The differences in the transfer frequency results obtained may be due to the initial sensitivity of the strains to the antibiotics tested—in our study, strains showing sensitivity to antibiotics carrying so-called silent antibiotic resistance genes were used, as in a previous study by the researchers [19]. In contrast, in the other study, the authors [18] only focused on resistant strains. Studies show that HPP, by affecting the structure of bacterial cell membranes, can promote increased cell permeability and potentially increase the frequency of gene transfer [38].

The use of genotypic analysis revealed differences in genetic structure between transconjugants subjected to different pressures. The literature indicates that HPP can cause changes in DNA structure and disruption of cellular processes, which translates into the observed cluster variation on the dendrogram (Figure 1) [17,38,39]. The dendrogram obtained by UPGMA based on ERIC-PCR profiles provides important information on the genetic similarity between *E. faecalis* strain JH2-2 and transconjugants obtained from donors subjected to different levels of pressure. This analysis not only allows us to understand the effect of pressure stress on the bacterial genotype in vitro but also provides important information on potential adaptation mechanisms under more complex environmental conditions, in situ. Statistical evidence further corroborates these findings. For example, Friedman’s ANOVA and Spearman correlation analyses demonstrated the genetic impact of HPP-induced stress on transconjugants, particularly at 400 MPa, where more pronounced genetic rearrangements and increased resistance gene transfer were observed. Control samples, representing transconjugants obtained from donors before HPP treatment, show a high degree of genetic similarity, confirming genome stability under conditions without pressure stress. Their close clustering on the dendrogram reflects the stable transfer of genetic material under laboratory conditions. Transconjugants exposed to 200 MPa pressure form a cluster separate from the control group, but show no significant differences, suggesting moderate genetic changes. This is probably due to the mild pressure stress, which induces adaptive responses but does not lead to drastic genetic rearrangements. Similar observations were made by Zarzecka et al. (2023) [18], who indicated that moderate pressure could lead to adaptations, but without significant changes in the bacterial genome compared to higher pressure. The most distinct cluster is formed by transconjugants subjected to 400 MPa pressure, which form a separated group on the dendrogram. This separation suggests that high pressure induces significant changes in the genetic material, probably related to genetic rearrangements or mutations. Similar phenomena have been described in the literature, where high pressure led to significant genome rearrangements, particularly in bacteria adapted to extreme environmental conditions [40]. These reactions may reflect intense selection for increased survival under extreme conditions.

Evidence from studies suggests that *L. monocytogenes* can acquire and potentially transfer antibiotic resistance genes in the food environment, either through conjugation, plasmids, transposons, or other gene transfer mechanisms, often involving different bacterial species [5,8,9,10,12,13,30]. This phenomenon, documented by the presence of such genes in different strains [14,20], highlights the importance of monitoring and controlling antibiotic-resistant *L. monocytogenes* in the food system to minimize health risks [29,41]. High-quality surveillance and an understanding of the ecological factors influencing resistance gene transfer are key to managing this risk, especially in the context of the potential spread of resistance through the food chain and its impact on listeriosis treatment [42,43,44].

## 4. Materials and Methods

### 4.1. Isolates Characterization

#### 4.1.1. Donor Isolates Characterization

For this study, six *L. monocytogenes* isolates characterized in earlier studies by the authors for the following: (I) antibiotic resistance characterization were used as donor strains [20]; (II) survival, recovery in vitro, and changes in antibiotic resistance under HPP including the occurrence and expression of genes encoding antibiotic resistance [8]. Isolates were from the collection of isolates of the Department of Food Microbiology, Meat Technology and Chemistry of the University of Warmia and Mazury in Olsztyn. The results were interpreted as Resistant (R), Intermediate (I), or Sensitive (S). Values were referenced by the European Committee on Antimicrobial Susceptibility Testing (EUCAST) [45] and the Clinical and Laboratory Standards Institute (CLSI) guidelines [46].

This study tested donor isolates to determine whether the identified antibiotic-resistance genes were chromosomally or plasmid-located. For this purpose, PCR and real-time PCR reactions were repeated for the resistance genes tested as in previous studies [16,20], using plasmid DNA as a template. According to the manufacturer’s instructions, Plasmid DNA was isolated from the strains using the Plasmid Mini DNA Isolation kit (A&A Biotechnology, Gdańsk Poland).

Briefly, the real-time PCR reaction was carried out in a 10 µL Master Mix reaction containing 5 µL of PowerUp SYBR Green Master Mix (Thermo Fischer Scientific, Waltham, MA, USA), 1 µL of forward and reverse primer (800 nM/μL), and 1 µL of extracted plasmid DNA, supplemented with ddH2O to final volume. Real-time PCR conditions were as follows: 50 °C for 2 min, 95 °C for 10 min (initial denaturation), 40 cycles of 95 °C for 15 s and 60 s at the annealing temperature specific for each gene analyzed (*tetA_1* (59.7 °C), *tetA_3* (58.1 °C), *tetC* (58.6 °C), *lin* (56.9 °C), and *fosX* (56.9 °C)). The specificity of the real-time PCR product was assessed by constructing a melting curve using a stepwise increase between 60 and 95 °C with 0.5 °C increments for 5 s.

The PCRs were performed using 2 μL of template plasmid DNA, 2 μL of each primer (100 pmol), and 12.5 μL of 2 × DreamTaq PCR Master Mix (Thermo Fisher Scientific, Waltham, MA, USA) in a total reaction volume of 25 μL. PCR conditions were as follows for the *Lde* gene (product size—1518 bp): initial denaturation for 3 min at 95 °C, followed by 1 min of denaturation at 95 °C, 45 s annealing at 45 °C, 1 min of extension at 72 °C for a total of 35 cycles and 5 min of final extension at 72 °C; and for *lnuA* gene (product size—323 bp): initial denaturation for 5 min at 94 °C, followed by 1 min of denaturation at 94 °C, 1 min annealing at 59 °C, 2 min of extension at 72 °C for a total of 35 cycles and 5 min of final extension at 72 °C.

A negative result (no band in the case of the PCR reaction and no increase in fluorescence in the case of real-time PCR) indicated chromosomal localization of the gene. This study only analyzed the transfer of plasmid-localized genes in the isolates. The characteristics of the donor isolates are shown in Table 7.

#### 4.1.2. Recipient Strain Characterization

*E. faecalis* JH2-2 strain (LMG 19456) was used as the recipient strain. This strain is plasmid-free and shows resistance to rifampicin (MIC > 256 μg/mL) and fusidic acid (MIC > 256 μg/mL). The strain was subjected to additional resistance characterization tests against the five antibiotics selected in this study (tetracycline (TET), lincomycin (LIN), ciprofloxacin (CIP), fosfomycin (FOS) and clindamycin (DA)) using the microdilution method, according to the previous study [16]. *E. faecalis* ATCC^®^ 29212 was used as the quality control organism.

Until proper analyses, the donor isolates and the recipient strain were stored in microbanks (Biomaxima, Lublin, Poland) at −80 °C. The analyzed isolates were cultured on Tryptic Soy Broth (TSB; Merck, Darmstadt, Germany) at 37 °C for 24 h. Each resulting colony was then streaked onto Tryptic Soy Agar (TSA; Merck, Darmstadt, Germany) and incubated under the same conditions. The resulting cultures were used for further analyses.

### 4.2. High-Pressure Processing

Each of the six *L. monocytogenes* isolates analyzed was subjected to the stress of two combinations of HPP treatment pressure and time. The conditions of the HPP were applied according to the conditions in previous studies [16]. The study used two pressure variants 200 MPa and 400 MPa for 5 minutes. Briefly, 10 mL of a 24 h culture of each isolate analyzed in Brain Heart Infusion broth (BHI; Merck, Darmstadt, Germany) (in vitro studies) and 10 mL of a 24 h culture of each isolate analyzed in carrot juice (Vital Fresh, Poland) (in situ studies) were transferred to a low-density polyethylene bottle (Kautex, Bonn, Germany) and subjected to HPP treatment in a solution of glycol and water (1:1, *v*/*v*), in a U4040 high-pressure single chamber (IWC PAN, Warsaw, Poland, Unipress Equipment Division) at a temperature of 20 ± 3 °C. The rate of pressure build-up was 300 MPa/min, while the pressure relief time was <5 s. Samples not treated with HPP were used as a control. For each isolate tested, tests were carried out in three independent replicates. After HPP treatment, the following steps of this study were carried out.

#### 4.2.1. Survival and Recovery Analysis After HPP Treatment In Vitro and In Situ

The survival rate and the possibility of recovery of the tested strains were checked in two variants—in vitro (in microbiological medium) and in situ (in food matrix—carrot juice), according to the methodology proposed in an earlier study [16]. The survival rate was tested using the plate count method with serial dilutions of isolates before and after HPP treatment and their inoculation on agar for *Listeria* according to Ottaviani and Agosti (ALOA) (Merck, Darmstadt, Germany). After 48 h incubation at 37 °C, colonies grown on the medium were counted and reported as the number of colony-forming units (CFU) per mL. Survival and recovery were tested in three, independent replicates.

When colonies were not detected on the plates using the plate count method (cell count below the detection level of the method), a cell recovery check was performed. The presence or absence of *L. monocytogenes* was monitored immediately after HPP treatment and during storage (at 1-day intervals until growth was detected, at 30 °C). Briefly, 1 mL of each bacterial culture was transferred into fresh TSB medium (Merck, Darmstadt, Germany) (for in vitro studies) or carrot juice (for in situ studies) and incubated at 30 °C until cell recovery. The suspension was inoculated onto TSA medium (Merck, Darmstadt, Germany), and after growth of each isolate was observed (in triplicate), TSA plates were incubated for 48 h at 30 °C. For growth, five random colonies were confirmed by culture on ALOA agar and incubation for 24 h at 37 °C. The experiment was repeated three times. After recovery, isolates were cultured on fresh TSB and immediately subjected to the next test steps.

#### 4.2.2. Change in Antibiotic Resistance

After applying HPP, MIC values were determined for five antibiotics: lincomycin, fosfomycin, tetracycline, ciprofloxacin, and clindamycin. Changes in MIC values were performed directly after stress treatment (200 MPa/5 min), and after the recovery of cells during storage (after exposure to 400 MPa/5 min). MIC values were determined using the microdilution method according to the previous study [16]. MIC values were read after a 20 h incubation at 35 ± 2 °C. The MIC value was the lowest antibiotic concentration at which no visible growth was observed in the microtiter plate wells. Values were referenced by the European Committee on Antimicrobial Susceptibility Testing (EUCAST) [45] and the Clinical and Laboratory Standards Institute (CLSI) guidelines [46]. Results were interpreted as susceptible (S), intermediate resistant (I), or resistant (R).

### 4.3. Transfer of Antibiotic Resistance Gene

In the study, the transfer of seven genes encoding resistance to five different antibiotics was analyzed—tetracycline (*tetA_1*, *tetA_3*, *tetC*), lincomycin (*lin*), ciprofloxacin (*Lde*), fosfomycin (*fosX*), and clindamycin (*lnuA*) in isolates possessing plasmid-encoded genes.

#### 4.3.1. In Vitro

The experiment was performed using the membrane filter method [47]. Strains after HPP in microbial medium (200 MPa/5 min), and strains after recovery in microbial medium (after 400 MPa/5 min) were used as donors, and *E. faecalis* strain JH2-2 was used as the recipient. Donor strains not exposed to HPP treatment were gene transfer controls. Gene transfer was performed according to the scheme presented by Wiśniewski et al. (2024) [19] with some modifications (Figure 2). For strains, regeneration was performed after HPP treatment at 400 MPa/5 min. After the recovery of the cells, gene transfer proceeded according to the mentioned scheme. Three types of solid microbial media were used in this study—the first for transconjugants (T-BHI), containing 100 µg/mL fusidic acid, 50 µg/mL rifampicin and the appropriate concentration of the antibiotic encoding by the gene for which the transfection was performed; second—for donor (TD-BHI), containing only the proper concentration of the antibiotic for which resistance is encoded by the gene for which the transfer was carried out; third—for recipients (B-BHI), containing 100 µg/mL fusidic acid and 50 µg/mL rifampicin. To exclude the possibility of transformation DNase I (ThermoFisher Scientific, Waltham, MA, USA) was added to the plates (100 µg/mL). Transfer results were reported and defined as the transfer frequency, i.e., the number of transconjugant colonies (CFU/mL) divided by the number of donor cells.

#### 4.3.2. In Situ

The possibility of gene transfer in a food matrix was tested using carrot juice (Vital Fresh, Poland). Strains after HPP treatment in carrot juice (200 MPa/5 min). Strains after regeneration in carrot juice (after 400 MPa/5 min) were used as donors, and *E. faecalis* strain JH2-2 was used as the recipient. Donor strains not exposed to HPP treatment were gene transfer controls. Gene transfer was carried out according to Figure 2, using tenfold dilution and inoculation into three previously characterized media: T-BHI, TD-BHI, and B-BHI.

#### 4.3.3. Characteristics of the Transconjugants

The transconjugants obtained were analyzed for resistance to the tested antibiotics (determination of MIC values), according to the methodology described earlier in this article (Section 4.2.2. Change in Antibiotic Resistance).

To confirm gene transfer, the isolation of genomic and plasmid DNA from the transconjugants obtained was carried out using the Genomic Mini kit (A&A Biotechnology, Gdańsk, Poland) and Plasmid Mini kit (A&A Biotechnology, Gdańsk, Poland), respectively, and PCR and real-time PCR reactions were performed for the resistance genes analyzed, following previously studies [16].

All transconjugants obtained were genotyped using molecular typing by enterobacterial repetitive intergenic consensus polymerase chain reaction (ERIC-PCR), according to Zarzecka et al. (2022) [31]. ERIC-PCR was performed in a 25 µL mixture (12.5 µL DreamTaq Green PCR Master Mix (2×) (ThermoFisher Scientific, Watham, MA, USA), 1.25 µL 10 pmol of each primer, 8.5 µL ddH2O and 1.5 µL DNA template), in a Mastercycler nexus GX2 thermocycler (Eppendorf, Germany). PCR reaction products were separated using electrophoresis in 1.5% agarose gels and imaged using the G-BOX F3 system (Syngene, UK).

For confirmation, the transconjugant fingerprints obtained were matched with those for the recipient strain. BioNumerics software version 7.6.3 (Applied Maths/bioMérieux, Sint-Martens-Latem, Belgium) was used for cluster analysis. Similarity distances between ERIC-PCR profiles were calculated using the Dice coefficient, and the dendrogram was based on an unbalanced clustering coefficient using the arithmetic mean method (UPGMA). Additionally, for confirmation, all transconjugant strains were genotyped to the species level using PCR and MALDI-TOF as described previously [48].

### 4.4. Statistical Analysis

A series of statistical tests was used to analyze the resistance gene transfer frequency data to assess the relationships and differences between different experimental conditions. The Shapiro–Wilk test was used to assess the normality of the distribution of resistance gene transfer frequency data in different experiments, such as control, at 200 MPa and recovery at 400 MPa. Since the data did not meet the normality assumption (*p* < 0.05), non-parametric tests were performed. Friedman’s ANOVA test was used to analyze differences in gene transfer frequencies between different pressure conditions and to examine the effect of pressure on resistance gene transfer. In addition, the Kruskal–Wallis test was used to compare the frequency of resistance gene transfer between different antibiotics under the same conditions. Finally, the Spearman correlation was used to assess the strength and direction of the relationship between pressure conditions and gene transfer frequency, examining how changes in pressure affect resistance gene transfer in the samples studied. The statistical analysis was conducted using PQStat Software v. 1.8.6 (Poznań, Poland). A significance threshold of *p* < 0.05 was assumed. Data are expressed as mean ± standard deviation (SD) unless otherwise stated.

## 5. Conclusions

The effect of high-pressure processing on antibiotic resistance gene transfer is complex and depends on specific bacterial strains, resistance genes, and environmental conditions. The present study confirms the findings on the role of HPP in increasing the transfer of ARGs in food-related environments and the potential of strains showing initial antibiotic susceptibility possessing so-called silent antibiotic resistance genes to transfer them. Statistical analysis confirmed significant differences in gene transfer frequencies under different pressure levels and environmental conditions. These findings demonstrate the intricate relationship between pressure conditions, bacterial behavior, and resistance gene transfer.

Our studies show that *L. monocytogenes* show varying resistance to high-pressure processing and are strongly dependent on environmental conditions, such as the composition and properties of the food matrix. The protective properties of food substrates, such as carrot juice, give the bacteria an advantage, allowing some strains to survive under conditions where their growth would be inhibited (in simpler media). This suggests that the interaction between food components and bacterial responses to stress significantly determines bacterial survival under extreme conditions. Furthermore, antibiotic resistance gene transfer between *L. monocytogenes* and *E. faecalis* is influenced by HPP, with higher pressure (400 MPa) generally promoting gene transfer of some resistance genes, such as those for tetracycline (*tet_A1*, *tet_A3*, *tetC*), both in vitro and in situ, while inhibiting the transfer of others (the *Lde* gene—the gene encoding ciprofloxacin resistance) under any conditions. The complexity of this process highlights the potential risk that food environments may pose as reservoirs of ARGs, with the possibility of further resistance transfer along the food chain exacerbated by food processing techniques, such as HPP. Pressure conditions, such as 200 MPa and 400 MPa, had varying effects on the transfer of different resistance genes depending on the antibiotic type and food matrix. This highlights the need for further studies to understand how HPP may influence the risk of resistance gene transfer in real-world food production environments.

Food matrices and related factors can significantly influence the transfer of antibiotic resistance genes. This requires a comprehensive approach to reducing antibiotic resistance in food production systems, carefully selecting processing and preservation methods. Understanding these interactions is key to developing effective strategies to combat the spread of antibiotic resistance in the food production chain.

Genetic analysis of the transconjugants showed that HPP can cause genotypic changes, especially at higher pressures, leading to distinct genetic rearrangements. This highlights the potential of HPP to influence the genomic stability of bacterial populations in food environments, contributing to adaptive responses under stress conditions. Future research should focus on fully understanding the long-term consequences of genomic rearrangements and developing strategies to mitigate potential risks associated with HPP-induced bacterial adaptations.

Our results suggest that although HPP effectively reduces bacterial populations, its effect on ARG transfer is variable. These findings highlight the need for further research into the use of HPP in food processing, particularly its impact on the spread of antibiotic resistance in the food industry. Effective monitoring and the development of new HPP control strategies will be key to preventing the spread of resistant bacterial strains, which will directly affect public safety and consumer health.

## Figures and Tables

**Figure 1 ijms-25-12964-f001:**
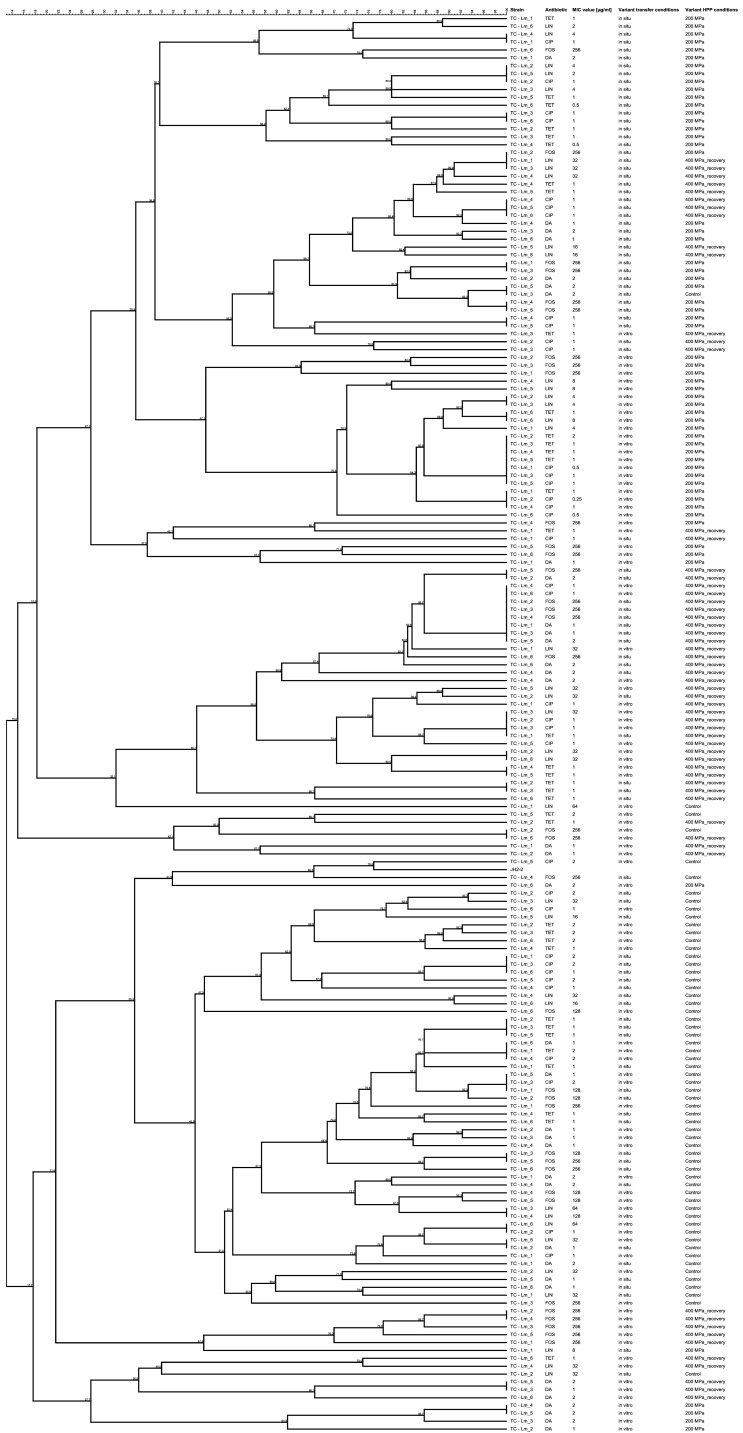
The UPGMA dendrogram obtained from ERIC-PCR profiles shows the restriction pattern similarity between *E. faecalis* JH2-2 and the obtained transconjugants. Abbreviations: TC—transconjugant. Control—control sample (transconjugant obtained from the strain before HPP treatment). 200 MPa—transconjugant obtained from the strain after exposure to 200 MPa pressure; 400 MPa—transconjugant obtained from the recovered strain after exposure to 400 MPa pressure.

**Figure 2 ijms-25-12964-f002:**
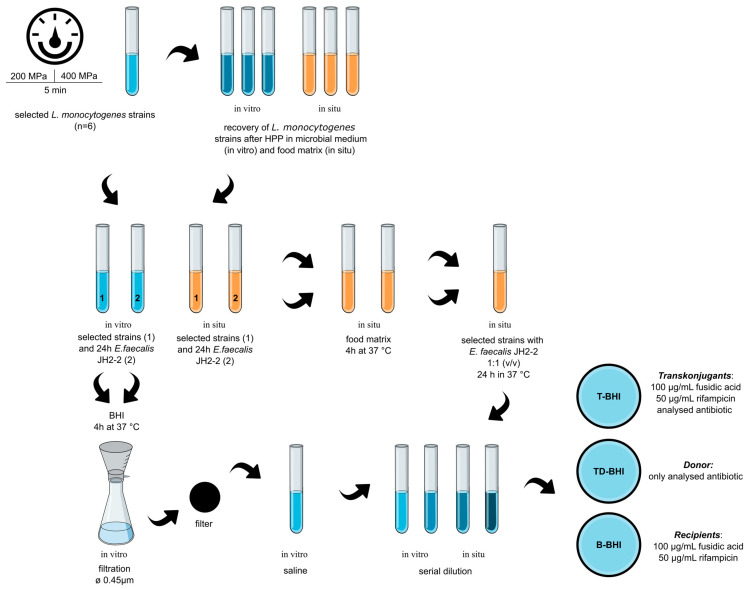
Diagram of transfer of antibiotic resistance genes in vitro and in situ.

**Table 1 ijms-25-12964-t001:** Plasmid localization of antibiotic resistance genes in analyzed *L. monocytogenes* strains.

	Antibiotic Resistance							
	TET			LIN	CIP	FOS	DA	
	*tetA_1*	*tetA_3*	*tetC*	*lin*	*Lde*	*fosX*	*lnuA*	*mefA*
**Lm_1**	+	+	+	+	+	+	+	ND
**Lm_2**	+	+	+	+	+	+	ND	+
**Lm_3**	+	+	+	+	+	+		ND
**Lm_4**	+	+	+	+	+	+		
**Lm_5**	+	+	+	+	+	+		
**Lm_6**	+	+	+	+	+	+	+	+

Abbreviation: TET—tetracycline, LIN—lincomycin, CIP—ciprofloxacin, FOS—fosfomycin, DA—clindamycin; ND—not detected; +—presence of a gene encoded on the plasmid.

**Table 2 ijms-25-12964-t002:** The viable cell count of *L. monocytogenes* isolates was analyzed in vitro.

	Control *	200 MPa *	400 MPa *	Recovery After 400 MPa
**Lm_1**	7.15 ± 0.22 × 10^9^	2.18 ± 0.15 × 10^9^	<10	Yes
**Lm_2**	2.70 ± 0.20 × 10^9^	1.44 ± 0.10 × 10^9^	<10	
**Lm_3**	1.91 ± 0.10 × 10^9^	1.64 ± 0.11 × 10^9^	<10	
**Lm_4**	2.26 ± 0.21 × 10^9^	1.87 ± 0.25 × 10^9^	<10	
**Lm_5**	2.68 ± 0.10 × 10^9^	1.35 ± 0.12 × 10^9^	<10	
**Lm_6**	2.08 ± 0.19 × 10^9^	1.77 ± 0.13 × 10^9^	<10	

Abbreviation: * colony forming units (CFU/mL); <10—number of bacteria below the detection limit; control—control sample (before stress treatment); 200 MPa—directly after exposure to 200 MPa pressure; 400 MPa—directly after exposure to 400 MPa pressure.

**Table 3 ijms-25-12964-t003:** The viable cell count of *L. monocytogenes* isolates was analyzed in situ.

	Control *	200 MPa *	400 MPa *	Recovery After 400 MPa
**Lm_1**	7.03 ± 0.12 × 10^8^	7.01 ± 0.10 × 10^8^	<10	Yes
**Lm_2**	7.53 ± 0.13 × 10^8^	7.41 ± 0.12 × 10^8^		
**Lm_3**	7.84 ± 0.13 × 10^8^	7.78 ± 0.13 × 10^8^		
**Lm_4**	8.52 ± 0.02 × 10^8^	8.34 ± 0.22 × 10^8^	2.00 ± 0.10 × 10^1^	NA
**Lm_5**	6.68 ± 0.18 × 10^8^	6.68 ± 0.13 × 10^8^	7.00 ± 0.20 × 10^1^	
**Lm_6**	6.01 ± 0.13 × 10^8^	5.97 ± 0.15 × 10^8^	3.00 ± 0.20 × 10^1^	

Abbreviation: * colony forming units (CFU/mL); <10—number of bacteria below the detection limit; NA—not applicable; control—control sample (before stress treatment); 200 MPa—directly after exposure to 200 MPa pressure; 400 MPa—directly after exposure to 400 MPa pressure.

**Table 4 ijms-25-12964-t004:** Results of gene transfer in microbial culture medium (in vitro) and food matrix (in situ) before and after exposure to HPP.

Antibiotic	TC	Microbial Culture Medium (In Vitro)		Food Matrix (In Situ)	
		200 MPa	Recovery After 400 MPa	200 MPa	Recovery After 400 MPa
**TET**	TC-Lm_1	** ↓ **	↑	** ↓ **	↑
	TC-Lm_2	** ↓ **	↑	↑	** ↓ **
	TC-Lm_3	** ↓ **	↑	↑	** ↓ **
	TC-Lm_4	↑	↑	** ↓ **	↑
	TC-Lm_5	** ↓ **	↑	** ↓ **	↑
	TC-Lm_6	** ↓ **	** ↓ **	↑	↑
**LIN**	TC-Lm_1	↑	↑	** ↓ **	↑
	TC-Lm_2	↑	** ↓ **	↑	↑
	TC-Lm_3	↑	↑	** ↓ **	↑
	TC-Lm_4	** ↓ **	↑	↑	↑
	TC-Lm_5	↑	↑	** ↓ **	** ↓ **
	TC-Lm_6	↑	↑	↑	** ↓ **
**CIP**	TC-Lm_1	-	** ↓ **	↑	↑
	TC-Lm_2	-	** ↓ **	↑	↑
	TC-Lm_3	** ↓ **	↑	↑	↑
	TC-Lm_4	** ↓ **	** ↓ **	↑	↑
	TC-Lm_5	** ↓ **	↑	↑	↑
	TC-Lm_6	↑	↑	↑	↑
**FOS**	TC-Lm_1	↑	↑	↑	↑
	TC-Lm_2	↑	↑	↑	↑
	TC-Lm_3	↑	↑	** ↓ **	** ↓ **
	TC-Lm_4	↑	↑	** ↓ **	** ↓ **
	TC-Lm_5	** ↓ **	↑	↑	** ↓ **
	TC-Lm_6	↑	** ↓ **	** ↓ **	** ↓ **
**DA**	TC-Lm_1	** ↓ **	↑	** ↓ **	** ↓ **
	TC-Lm_2	** ↓ **	** ↓ **	↑	** ↓ **
	TC-Lm_3	** ↓ **	** ↓ **	↑	↑
	TC-Lm_4	↑	** ↓ **	** ↓ **	↑
	TC-Lm_5	** ↓ **	** ↓ **	** ↓ **	↑
	TC-Lm_6	↑	** ↓ **	↑	↑

Abbreviations: **↓**—a decrease in the frequency of gene transfer compared to the control value; ↑—increase in gene transfer frequency compared to control value; -—no decrease in the frequency of gene transfer compared to the control value; TC—transconjugant; TC/R—transfer rate, TET—tetracycline, LIN—lincomycin, CIP—ciprofloxacin, FOS—fosfomycin, DA—clindamycin; 200 MPa—transconjugant obtained from the strain after exposure to 200 MPa pressure; 400 MPa—transconjugant obtained from the recovered strain after exposure to 400 MPa pressure.

**Table 5 ijms-25-12964-t005:** Presence of antibiotic resistance genes in transconjugants after gene transfer in microbial culture medium (in vitro) and food matrix (in situ) before and after exposure to HPP.

Antibiotic	TC	Microbial Culture Medium (In Vitro)			Food Matrix (In Situ)		
		Control	200 MPa	Recovery After 400 MPa	Control	200 MPa	Recovery After 400 MPa
**TET**	TC-Lm_1	*tet_A1*, *tet_A3*, *tetC*	*tet_A1*, *tet_A3 ^p^*	*tet_A1*, *tet_A3*, *tetC*	*tet_A1*, *tet_A3*	*tet_A1*, *tet_A3*	*tet_A1 ^g^*, *tet_A3*
	TC-Lm_2				*tet_A1*, *tet_A3 ^p^*		
	TC-Lm_3		*tet_A1*, *tet_A3*	*tet_A1*, *tet_A3 ^p^*, *tetC*			
	TC-Lm_4	*tet_A1*, *tet_A3*, *tetC ^p^*		*tet_A1*, *tet_A3*, *tetC*		*tet_A1 ^p^*, *tet_A3*	*tet_A1*, *tet_A3*
	TC-Lm_5	*tet_A1*, *tet_A3*, *tetC*	*tet_A1*, *tet_A3 ^p^*				
	TC-Lm_6	*tet_A1*, *tet_A3*, *tetC ^p^*				*tet_A1 ^p^*, *tet_A3 ^p^*	
**LIN**	TC-Lm_1	*lin*	*lin*	-	*lin*	*lin*	*lin*
	TC-Lm_2						
	TC-Lm_3						
	TC-Lm_4						
	TC-Lm_5						
	TC-Lm_6						
**CIP**	TC-Lm_1	-					
	TC-Lm_2						
	TC-Lm_3						
	TC-Lm_4						
	TC-Lm_5						
	TC-Lm_6						
**FOS**	TC-Lm_1	*fosX ^p^*	-	-	*fosX ^p^*	-	-
	TC-Lm_2		*fosX ^p^*	*fosX*		*fosX ^p^*	
	TC-Lm_3		*fosX*				
	TC-Lm_4		*fosX ^p^*				
	TC-Lm_5						
	TC-Lm_6						
**DA**	TC-Lm_1	*lnuA ^p^*	-	-	*lnuA ^p^*	-	-
	TC-Lm_2	NA					
	TC-Lm_3						
	TC-Lm_4						
	TC-Lm_5						
	TC-Lm_6	*lnuA ^p^*	-	-	*lnuA ^p^*	-	-

Abbreviations: TC—transconjugant; TET—tetracycline, LIN—lincomycin, CIP—ciprofloxacin, FOS—fosfomycin, DA—clindamycin. Control—control sample (transconjugant obtained from the strain before HPP treatment); 200 MPa—transconjugant obtained from the strain after exposure to 200 MPa pressure; 400 MPa—transconjugant obtained from the recovered strain after exposure to 400 MPa pressure. *^g^*—gene present only in genomic DNA; *^p^*—gene present only in plasmid DNA; no superscript (*^p/g^*)—gene present in both genomic and plasmid DNA; NA—not applicable.

**Table 6 ijms-25-12964-t006:** Results of changes in the values of minimum inhibitory concentrations (MICs) in microbial culture medium (in vitro) and food matrix (in situ) before and after exposure to HPP.

Antibiotic	TC	Microbial Culture Medium (In Vitro)			Food Matrix (In Situ)		
		Control	200 MPa	Recovery After 400 MPa	Control	200 MPa	Recovery After 400 MPa
		MIC’s Values [µg/mL]					
**TET**	TC-Lm_1	2 (×8↑)	1 (×4↑)	1 (×4↑)	1 (×4↑)	1 (×4↑)	1 (×4↑)
	TC-Lm_2	2 (×16↑)	2 (×16↑)	1 (×8↑)	1 (×8↑)	1 (×8↑)	1 (×8↑)
	TC-Lm_3	2 (×8↑)	1 (×4↑)	1 (×4↑)	1 (×4↑)	1 (×4↑)	1 (×4↑)
	TC-Lm_4	1 (×8↑)	1 (×8↑)	1 (×8↑)	1 (×8↑)	0.50 (×4↑)	1 (×8↑)
	TC-Lm_5	2 (×4↑)	1 (×2↑)	1 (×2↑)	1 (×2↑)	1 (×2↑)	1 (×2↑)
	TC-Lm_6	2 (×4↑)	1 (×2↑)	1 (×2↑)	1 (×2↑)	0.50 (-)	1 (×2↑)
**LIN**	TC-Lm_1	64 (×4↑)	4 (×4↓)	32 (×2↑)	32 (×2↑)	8 (×2↓)	32 (×2↑)
	TC-Lm_2	32 (×4↑)	4 (×2↓)	32 (×4↑)	32 (×4↑)	4 (×2↓)	32 (×4↑)
	TC-Lm_3	64 (×128↑)	4 (×8↑)	32 (×64↑)	32 (×64↑)	4 (×8↑)	32 (×64↑)
	TC-Lm_4	128 (×128↑)	8 (×8↑)	32 (×32↑)	32 (×32↑)	4 (×4↑)	32 (×32↑)
	TC-Lm_5	32 (×64↑)	8 (×16↑)	32 (×64↑)	16 (×32↑)	2 (×4↑)	16 (×32↑)
	TC-Lm_6	64 (×128↑)	8 (×16↑)	32 (×64↑)	16 (×32↑)	2 (×4↑)	16 (×32↑)
**CIP**	TC-Lm_1	1 (×8↑)	0.50 (×4↑)	1 (×8↑)	2 (×16↑)	1 (×8↑)	1 (×8↑)
	TC-Lm_2	1 (×8↑)	0.25 (×2↑)	1 (×8↑)	2 (×16↑)	1 (×8↑)	1 (×8↑)
	TC-Lm_3	2 (×16↑)	1 (×8↑)	1 (×8↑)	2 (×16↑)	1 (×8↑)	1 (×8↑)
	TC-Lm_4	2 (×16↑)	1 (×8↑)	1 (×8↑)	1 (×8↑)	1 (×8↑)	1 (×8↑)
	TC-Lm_5	2 (×16↑)	1 (×8↑)	1 (×8↑)	2 (×16↑)	1 (×8↑)	1 (×8↑)
	TC-Lm_6	1 (×4↑)	0.50 (×2↑)	1 (×4↑)	1 (×4↑)	1 (×4↑)	1 (×4↑)
**FOS**	TC-Lm_1	256 (-)	256 (-)	256 (-)	128 (×2↓)	256 (-)	256 (-)
	TC-Lm_2	256 (-)	256 (-)	256 (-)	128 (×2↓)	256 (-)	256 (-)
	TC-Lm_3	256 (×64↑)	256 (×64↑)	256 (×64↑)	128 (×32↑)	256 (×64↑)	256 (×64↑)
	TC-Lm_4	128 (×64↑)	256 (×128↑)	256 (×128↑)	256 (×128↑)	256 (×128↑)	256 (×128↑)
	TC-Lm_5	128 (×8↑)	256 (×16↑)	256 (×16↑)	256 (×16↑)	256 (×16↑)	256 (×16↑)
	TC-Lm_6	128 (×4↑)	256 (×8↑)	256 (×8↑)	256 (×8↑)	256 (×8↑)	256 (×8↑)
**DA**	TC-Lm_1	2 (-)	1 (×2↓)	1 (×2↓)	2 (-)	2 (-)	1 (×2↓)
	TC-Lm_2	1 (×4↑)	1 (×4↑)	1 (×4↑)	1 (×4↑)	2 (×8↑)	2 (×8↑)
	TC-Lm_3	1 (-)	2 (×2↑)	1 (-)	2 (×2↑)	2 (×2↑)	1 (-)
	TC-Lm_4	1 (×8↑)	2 (×16↑)	2 (×16↑)	2 (×16↑)	1 (×8↑)	2 (×16↑)
	TC-Lm_5	1 (×1.5↓)	2 (×1.3↑)	2 (×1.3↑)	1 (×1.5↓)	2 (×1.3↑)	2 (×1.3↑)
	TC-Lm_6	1 (×2↑)	2 (×4↑)	2 (×4↑)	1 (×2↑)	1 (×2↑)	2 (×4↑)

Abbreviations: TC—transconjugant; TET—tetracycline, LIN—lincomycin, CIP—ciprofloxacin, FOS—fosfomycin, DA—clindamycin; Control—control sample (transconjugant obtained from the strain before HPP treatment); 200 MPa—transconjugant obtained from the strain after exposure to 200 MPa pressure; 400 MPa—transconjugant obtained from the recovered strain after exposure to 400 MPa pressure. Values in parentheses—indicate how many times the MIC value increased (↑), decreased (↓), or remained unchanged (-) in transconjugants compared to the donor strains.

**Table 7 ijms-25-12964-t007:** Characteristics of *L. monocytogenes* donor isolates used in this study.

Strain		Serotype	Isolation Source	Antibiotic MICs [µg/mL]					Antibiotic Resistance Genes
				TET	LIN	CIP	FOS	DA	
**Lm_1**	168	1/2c	Floor drain	0.25 (S)	16 (R)	0.125 (S)	256 (R)	2 (I)	*tetA_1*, *tetA_3*, *tetC*, *lin*, *Lde*, *fosX*, *lnuA*,
**Lm_2**	165		Floor drain	0.125 (S)	8 (R)	0.125 (S)	>256 (R)	0.25 (S)	*tetA_1*, *tetA_3*, *tetC*, *lin*, *Lde*, *fosX*, *mefA*
**Lm_3**	177		Production line	0.25 (S)	0.5 (S)	0.125 (S)	4 (S)	1 (I)	*tetA_1*, *tetA_3*, *tetC*, *lin*, *Lde*, *fosX*,
**Lm_4**	92	1/2a	Juice	<0.125 (S)	1 (I)	0.125 (S)	2 (S)	0.125 (S)	*tetA_1*, *tetA_3*, *tetC*, *lin*, *Lde*, *fosX*,
**Lm_5**	167		Floor drain	0.5 (S)	0.5 (S)	0.125 (S)	16 (S)	1.5 (I)	*tetA_1*, *tetA_3*, *tetC*, *lin*, *Lde*, *fosX*,
**Lm_6**	148		Frozen vegetables	0.5 (S)	0.5 (S)	0.25 (S)	32 (S)	0.5 (S)	*tetA_1*, *tetA_3*, *tetC*, *lin*, *Lde*, *fosX*, *mefA*, *lnuA*

Abbreviation: TET—tetracycline, LIN—lincomycin, CIP—ciprofloxacin, FOS—fosfomycin, DA—clindamycin; R—resistance, I—intermediate resistance, S—susceptible.

## Data Availability

The data presented in this study are available in this article and Appendix A.

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
