# Peer review of "High-Pressure Processing Influences Antibiotic Resistance Gene Transfer in Listeria monocytogenes Isolated from Food and Processing Environments"

_ijms, 2024, doi:10.3390/ijms252312964_

Round 1
Reviewer 1 Report
Comments and Suggestions for Authors
The manuscript written by Patryk WiÅ›niewski, Wioleta ChajÄ™cka-Wierzchowska and Anna Zadernowska, „High-Pressure Processing Influences Antibiotic Resistance 2 Gene Transfer in Listeria monocytogenes Isolated from Food 3 and Processing Environments”, is a very well-written manuscript with important information about the transmission of resistance genes among bacteria.
I encourage its publication after minor corrections outlined below:
L 44 The authors mention” phenomenon observed in various studies,” and there is only one citation at the end of the paragraph, I suggest adding more references to this paragraph.
L93: In table 1: Why was DA applicable? for Lm_1 and Lm_6 but not the other isolates. I think the description „NA – not applicable” should be changed with something more appropriate like „not detected”
Reviewer 2 Report
Comments and Suggestions for Authors
Major points
Results
- Survival and Recovery Analysis – add a Table with the survival rates at both pressures
- Survival and Recovery Analysis – 1-2 images with the colony aspect on ALOA agar would improve the quality of the manuscript
- Table 2 needs to be redone (simplified/abbreviated) as it is exceedingly difficult to read and interpret
- 2.3.1. Gene transfer frequency – a further simplified table should be added that will make the interpretation easier
|
|
In vitro |
In situ carrot juice |
||
|
|
200 MPa |
Recovery af- ter 400 MPa |
200 MPa |
Recovery af- ter 400 MPa |
TET |
TC-Lm_1 |
- / ↑ / ↓ |
- / ↑ / ↓ |
- / ↑ / ↓ |
- / ↑ / ↓ |
|
TC-Lm_2 |
- / ↑ / ↓ |
- / ↑ / ↓ |
- / ↑ / ↓ |
- / ↑ / ↓ |
|
TC-Lm_3 |
- / ↑ / ↓ |
- / ↑ / ↓ |
- / ↑ / ↓ |
- / ↑ / ↓ |
|
|
|
|
|
|
- no significant change after exposure; ↑ significant increase in gene transfer; ↓ significant decrease in gene transfer
- 2.3.1. Gene transfer frequency - why are there no statistical tests applied to compare the means resulting from the gene transfer?
- 2.3.1. Gene transfer frequency – similarly the text requires further work in order to improve readability
- Table 3 – replace * and ** with a superscript that will allow an easier interpretation of the results such as g (genomic) and p (plasmid)
- Line 190 – “the tetC gene was not found in any of the transconjugants analysed, indicating that transfer of this gene was not possible.” This is partially true – the tetC gene was found in both control and after 400 MPa in vitro, indicating that the gene can be transferred under certain conditions. Specify that the transfer of the gene was not possible in situ
- 2.3.3. Changes in the Values of Minimum Inhibitory Concentrations – add a table with the results of the MIC values
Discussion
- the discussion section reiterates the same ideas multiple times – an effort should be made to rewrite the discussion section to better compare the obtained results with the existing data in the literature
Methods
- Table 4 – by which standard were the isolates characterised as S,R,I?
- 4.1.1 Donor isolates characterisation – expand this section to contain all details regarding the PCR procedure
- 4.3 Transfer of Antibiotic Resistance Gene – the methods regarding the testing of the transfer of antibiotic resistance genes can be further improved and clarified (e.g., add the antibiotics to the different medium types in Figure 2)
Conclusions
- the conclusions should also contain a take home message – the transfer of which genes and under which conditions were observed in this study
Minor points
- Introduction – add some examples of harsh food environments in which Listeria can survive; furthermore add some context to the contamination of the food chain with Listeria
- Introduction – add some data regarding the epidemiology of antibiotic resistance in Listeria
- Introduction – add some context to HPP – how often is it used, types of food, common pressure etc
- Line 61-64 – in which bacteria was this observed?
- Line 406 – add some minimal details regarding E. faecalis ATCC® 29212
Comments on the Quality of English LanguageMinor to moderate editing required
Round 2
Reviewer 2 Report
Comments and Suggestions for Authors
The manuscript has been visibly improved.
However, I would suggest a few additional minor changes:
- Line 422-423 - "It is important to observe that in our study, strains showing initial sensitivity to antibiotics carrying so-called antibiotic resistance genes were also capable of gene transfer." - should this not be silent antibiotic resistance genes?
- In the case of Table 4 a few percentages of increases/ no changes/ decreases in MIC values would be helpful in providing an overview of the results
